# Physico-Chemical Soil Properties Affected by Invasive Plants in Southwest Germany (Rhineland-Palatinate)—A Case Study

**Jellian Jamin** [1,*], **Dörte Diehl** [1], **Michele Meyer** [2], **Jan David** [1], **Gabriele Ellen Schaumann** [1] **and Christian Buchmann** [1,*]

1   iES Landau, Group of Environmental and Soil Chemistry, Institute for Environmental Sciences, University of Koblenz-Landau, Fortstraße 7, 76829 Landau, Germany
2   Helmholtz-Zentrum für Umweltforschung, 04318 Leipzig, Germany
\*   Correspondence: jamin@uni-landau.de (J.J.); buchmann@uni-landau.de (C.B.)

**Abstract:** The invasive plant species *Impatiens glandulifera* native to Asia mainly occupies European riparian ecosystems. It is still unclear to which extent this invasive plant can alter physico-chemical soil properties in terms of carbon turnover, microstructural stability and soil hydraulic properties threatening native plant species, here represented by *Urtica dioica.* Soil samples were collected from three sites in the Palatine forest near the river Queich, including bare soil (Control), or soil within dense stands of either *I. glandulifera* or *U. dioica* with similar texture. Basic soil parameters including SOM content and quality were analyzed. SOM is known to impact soil microstructural stability and soil hydraulic properties. We therefore assessed microstructural stability, the pore size distribution and the wettability. Our results implied more recalcitrant SOM for soil colonized by *U. dioca* including a lower pH. For soil colonized by *I. glandulifera* less recalcitrant SOM was detected indicating a reduced degradation which is likely given due to lignin as a predominant component in the plant biomass of *I. glandulifera* Soil microstructural stability was higher for soil colonized by the invader showing a slight increase with soil depth, due to higher SOM content. All in all, this case study indicates that *I. glandulifera* most likely affects the soil microbiome while basic soil parameters, soil hydraulic properties, wettability and soil microstructural stability showed no significant effect.

**Keywords:** invasive plants; soil hydraulic properties; DSC-TGA; rheology



## 1. Introduction

Riparian zones face dynamic challenges mainly due to changes in the surrounding ecosystem by natural and anthropogenic drivers [1], which include pollutants and invasive species [2,3]. Along rivers, their seeds or tissues are carried passively transported supporting their quick dispersal and colonization of riparian ecosystems after flooding [4]. Once established invasive plant species may modulate biogeochemical cycles [5] or show a quick growth [6] leading to a competitive advantage relative to native species. A prominent example is the Himalayan balsam (*Impatiens glandulifera* ROYLE), an annual plant native to India [7], superseding native plant populations [8]. While *Impatiens glandulifera* adds to local biodiversity and provides a food to insects [9,10], it promotes soil erosion due to shallow rooting [7]. In fact, *I. glandulifera* has a short, thick and tapered root system [11]. It's native competitor [12] amongst others the common nettle (i.e., *Urtica dioica* L.), has rhizomes. They are interconnected by long and intertwined fine roots spreading laterally through the rhizosphere [13]. Fine root hairs extend along the roots and away from the vegetation node [14]. Moreover, *I. glandulifera* most likely impacts the terrestrial soil carbon cycle by overall rapid growth [15]. *I. glandulifera* reaches heights of up to 2 m within a few weeks [4] producing large amounts of above-ground biomass rapidly [15,16] compared to slower growing, perennial European plants like the common nettle (i.e., *Urtica dioica* L.). Various studies have addressed the impact of *Impatiens glandulifera* for ecosystems [6,8,10,11] mainly focusing on ecological issues [9,17]. Insights on fundamental effects

on physico-chemical soil properties and mechanisms are still lacking [18]. The success of *I. glandulifera* has been observed starting from the 1990 years, describing botanical characteristics [4,19]. However, Ehrenfeld et al. [20] suggested in 2001 that invasive colonization might modulate biogeochemical cycles and Prescott et al. emphasized the impacts on litter decomposition implying changes in SOM dynamics [21]. SOM quantity and quality affects different soil properties, especially in terms of microstructural stability [22] and soil hydraulic properties such as wettability [23,24] and water binding in pore spaces [25]. The quality and quantity of SOM was assessed via Thermogravimetric analysis providing also the degree of metabolism expressed as thermal stability of various SOM fractions [26]. Wettability was tested using Optical Contact Angle tensiometry providing the Contact angle. Water binding was assessed using 1H-NMR relaxometry providing the pore size distribution and microstructural stability was tested using soil rheology in order to obtain the particle-particle interaction.

With the differing root systems (lateral vs. tapered) and the different vegetation patterns (perennial and slow vs. annual and fast), we came to the following hypotheses:

(I) We expected an enhanced soil organic matter (SOM) input for soil colonized by *I. glandulifera* due to its annual life cycle and the rapid growth compared to U. dioica with decreasing SOM content for deeper layers of soil.

(II) Addressing the root systems, we hypothesize that soil colonized by *U. dioica* exhibits a higher microstructural stability homogeneously distributed in different soil depth where the lateral, intertwined root acts as mechanical binding agent. In contrast, we expected a lower microstructural stability for soil colonized by *I. glandulifera* which decreases with soil depth due to the tapered root causing cracks in the soil microstructure.

Soil hydraulic properties are known, to depend on SOM content due to swelling upon water contact and hydration, leading to a shift in the pore size distribution (PSD) [25,27,28] from macropores towards micropores providing information on water retention under flooded conditions. Thus, we (III) hypothesized that the PSD of soil colonized by *I. glandulifera* is shifted to relatively more large pores with a simultaneous relative reduction in soil micropores with respect to the control soil. Since we expected lower SOM contents for soil colonized by *U. dioica* and the control soils, their PSD is expected to be shifted towards smaller pore sizes compared to *I. glandulifera*. Such differences in PSD can be obtained from 1H-NMR relaxometry measurements of undisturbed water saturated soil samples. During $^1$H-NMR relaxometry protons are exposed to a magnetic field, where protons can align to the angle of the applied magnetic field. A radio frequency pulse, causes an excitation of the protons which then return to equilibrium [29]. The decay of the magnetization is then measured depending on time and provides the volume of water located in different pores, since the so called relaxation time depends on the physical environment of the protons [30].

Regarding the wettability, we hypothesized (IV) that soil colonized by *I. glandulifera* shows a pronounced hydrophobicity due to higher SOM input compared to soil colonized by *U. dioica* or bare soil (control).

## 2. Materials and Methods

### 2.1. Study Site

Samples in the riparian zone of the river Queich in Rhineland-Palatinate (coordinates: 49.199787; 8.098031) in Germany were collected. An annual precipitation of 686 L/m$^2$ and an average temperature of 11.0 °C was reported for Rhineland palatinate [31]. The transects were selected depending on their vegetation status in a line of 500 m along the river bank to obtain soil with similar soil texture. The transects were either highly populated by the invasive plant species *Impatiens glandulifera*, by the native plant species *Urtica dioica* or with no vegetation as control (Figure 1).

The samples were collected in September 2020 after the main growing season within the stands. Five replicates for each vegetation type and each soil depth were collected with cylindrical containers (5 cm height, 3.5 cm diameter) and air-dried at room temperature in prior to analysis. This depth was considered, since *U. dioica* is known to root in topsoil [13]

and we additionally wanted to gain an insight on degradation dynamics, entering soil from aboveground-plant litter. From each vegetation cover (*I. glandulifera, U. dioica*) and the control and regarding each depth (surface = 0–2 cm, bulk = 2–5 cm) sampling was achieved in five-fold replication. The samples were stored in the same cylindrical containers with the purpose of maintaining the microstructure. The texture of the soil is categorized as silt loam (Table 1).

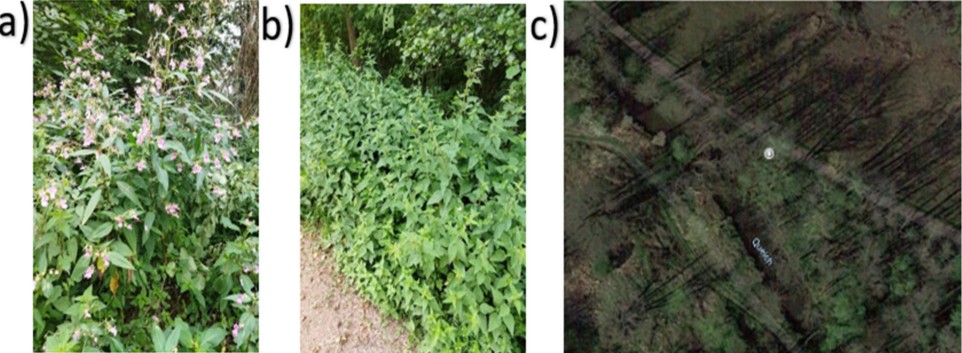

**Figure 1.** Impressions from the study sites of the riparian area during the vegetation season in August 2020 and the investigated stands (**a**) *Impatiens glandulifera* stand, (**b**) *Urtica dioica* stand, (**c**) satellite image from the respective site: shorturl.at/bpRT7 (accessed on 9 August 2022).

**Table 1.** Determined selected physico-chemical parameters of the investigated soil (dry mass basis).

| Parameter | |
| --- | --- |
| Texture | Loam silt |
| Sand (%) | 21.83 |
| Silt (%) | 68.80 |
| Clay (%) | 9.37 |
| Soil unit (WRB) [32] | gleysol |
| WHC$_{max}$ (g kg$^{-1}$) | 681 |

### *2.2. Soil Preparation and Characterization*

The dried samples were water-saturated over 7 days to their maximum water-holding capacity using a sandbox (09.01 Sandbox, Eijkelkamp, Zeitz, Germany) prior to further analysis to obtain all soil pores in the course of the $^1$H-NMR relaxometry experiments and minimize the effect of water menisci on the rheological measurements. The soil samples were subdivided into 'surface' (0–2 cm) and 'bulk' (2–5 cm) layer. Each soil parameter was determined for these respective two layers to assess depth-dependent differences and changes of the various soil properties. In the next section soil colonized by *U. dioica* will be referred to as *Urtica;* soil colonized by *I. glandulifera* as *Impatiens* and bare soil as control. Water content and dry mass of the samples were determined gravimetrically according to DIN EN 15934:2012-11. The grain size distribution was measured according to DIN ISO 17892-4:2017-04 for classification of soil texture. After $^1$H-NMR- and rheology experiments the samples were air-dried again and homogenized. Soil pH was measured in a 0.01 M CaCl$_2$-solution according to DIN EN 15933:2012-11 while electric conductivity was determined in a suspension in demineralized water according DIN CEN/TS 15937:2013-08.

### *2.3. $^1$H-NMR Relaxometry*

A Bruker Minispec MQ relaxometer (Bruker, Karlsruhe, Germany) was used to obtain the information on water retention of the samples at magnetic field strength of 0.176 T (proton Larmor frequency of 7.5 MHz). A Carr-Purcell-Meiboom-Gill (CPMG) pulse sequence and the corresponding relaxation rate of the water protons in the samples was applied [33]. Echo time and the recycle delay were kept constant at 300 μs and 10 s, respectively. Number of echoes, scans and gains were adapted for each sample individually.

Then, T2 was measured from both layers of each sample. The obtained decay curves were transformed with a MATLAB program applying a Butler, Reeds and Dawson (BRD) algorithm [29] reducing the $T_2$ measurement and the resulting relaxation time distribution (RTD) 200 points. From the RTD, the pore size distribution (PSD) was determined using the common calibration curve described by Meyer et al. [25].

### 2.4. Soil Rheometry

Soil rheometry assess the viscoelastic behavior of soil, where the sample is placed between two parallel plates. The lower plate is fixed and the upper plate is mobile applying vertical stress on the sample [22]. Soil microstructural stability of both bulk and surface layers was assessed via amplitude sweep tests according to Buchmann et.al [34] using an Anton Paar MCR 102 rheometer (Anton Paar, Ostfildern, Germany). Layers of approximately 3-4 mm were cut from the cylinders and placed on a parallel-plate measuring device. The transfer of the layers from the cylindrical container onto the fixed plate were achieved with minimal pressure to keep the structure as intact as possible. The setting for the deformation parameter $\gamma$ started from 1% and rose to 10% applying a logarithmic scale keeping a constant frequency of 0.5 Hz and 30 measurements points. Preliminary studies indicated the applicability of this range for obtaining the yield point. The temperature during the measurement was constantly kept at 20 °C by a Peltier unit. The applied force (vertical stress) was set below 1 N during the measurement. To avoid complete water evaporation from the samples the maximum duration of each measurement was set to 10 min. The following parameters were classified according to Holthusen et al. [35]. The shear stress at the yield point ($\tau_{YP}$), indicating the point of flow/creeping at which plastic deformation starts, the strain corresponding to the maximum shear stress ($\tau_{max}$) expressed as $\gamma_{(\tau max)}$ was described to be correlated with SOM content providing additional information on microstructural stability caused by soil particle interaction with SOM [34].

### 2.5. Determination of Dynamic Contact Angle Tensiometry (DCAT)

Dynamic contact angle measurements were executed on a Video-Based Optical Contact Angle Measuring Device (OCA15Pro, Data Physics, Filderstadt, Germany). In this study, a sessile drop method according to Bachmann et al. [36] was used to assess the wettability of the soil particles. For this, the air-dried and homogenized soil was piled with a spatula on an adhesive double-sided tape, which was fixed on a microscope glass slide and slightly pressed on the tape to form a fixed mono-particle-layer. Excess particles were removed by gently knocking on the glass slide. A droplet volume of 5 µL was placed on the particle layer and the drop formation recorded by a high-speed video camera, which allows the evaluation of the contact angle as a function of spreading at any time using the SCA 20 software (Data Physics, Filderstadt, Germany) according to Bachmann et al. [37].

### 2.6. Determination of SOM Content and Recalcitrance

The soil samples were air-dried and ball-milled prior to analysis using a simultaneous TG-DTA/DSC Apparatus STA 449 F3 Jupiter (NETZSCH, Selb, Germany) within a temperature range between 30–1000 °C in dynamic air with a flow rate of 100 mL min$^{-1}$ and a heating rate of 10 Kmin$^{-1}$. Sample masses of approximately 20 mg were weighed in $Al_2O_3$ crucibles [38]. A crucible with calcium-oxalate with approximately 10 mg was used as a reference. The resulting TG curve plots the mass loss or the loss of ignition (LOI) over time with increasing temperature. Energetic characteristics, i.e., the combustion energies ($\Delta H_{SOM}$ and $\Delta H_{lab.SOM}$) of the respective fractions are obtained by integration of the DSC curve versus time. Therefore, the base line was corrected to zero between 180 to 600 °C. The DSC curve revealed two events, associated with the combustion of a thermolabile organic matter and a thermostable organic matter fraction and a respective mass loss on the TG curve. These events appear as an exothermic combustion, occurring as peaks in the DSC curve between 200 and 600 °C. The total amount of SOM is related to the dry mass after evaporation of water at a temperature ranging between 100–200 °C and contains the sum

of the labile and the stable fraction SOM fraction [39]. Another quality index from the DSC curves is given by the temperature $T_{50}$ at which 50% of the organic matter is combusted providing information on the thermal recalcitrance of soil organic matter [40].

### 2.7. Data Analysis

Five replicates from two depths (surface and bulk) and three types (*U. dioica, I. glandulifera*, and bare soil as control) resulted in 30 samples in total. Since the requirements for normal distribution were not met, Kruskal–Wallis tests as a non-parametric variance analysis were applied [41] to detect significant effects of vegetation and depth on the obtained parameter. Effects were reported as significant at $p < 0.05$. Outliers were considered as well, since we wanted to evaluate the variation caused by the vegetation cover. A following post hoc Dunn's test using the Bonferroni method was conducted for a pairwise comparison to differentiate which groups were significant different from others [42]. Principal Component Analysis (PCA) was applied to detect relationships between variables. All statistical analysis was performed using Python 3.7 including Scipy.stats and sklearn packages. Mean values and standard errors of the measurements can be obtained from the Supporting Information (see Supplementary Material Table S1).

### 3. Results

#### 3.1. SOM Thermal Properties

Total of soil organic matter content (SOM) was not significantly different ($p > 0.05$) between soil samples neither at different depths nor when rooted by the different plants (Figure 2a). SOM content of soil colonized by *I. glandulifera* decreased with soil depth ranging between 7–14%. Soil colonized by *U. dioica* ranged between 7 and 12% and decreased with soil depth as well. Bare soil (Control) showed the lowest SOM contents ranging between 7–8%. No significant differences between the labile SOM fraction were detected ($p > 0.05$) between soil samples neither of at different depths nor when rooted by the different plants (Figure 2b).

The highest values for the labile SOM fractions were detected for soil colonized by *I. glandulifera* which decreased with soil depth as well and ranged between 3–7% followed by *U. dioica* ranging between 2–5.5% decreasing with soil depth. Bare soil (Control) showed contents of labile organic matter ranging between 3–5% with a slight increase with soil depth.

The respective combustion enthalpies for total and labile SOM range between −0.5 to −2.0 kJ·g$^{-1}$. No significant differences ($p > 0.05$) were detected between the combustion enthalpies, neither for the total nor for the labile organic matter fraction. Only the bulk layers of soil colonized by *I. glandulifera* showed slightly lower combustion enthalpies ranging from −0.5 to −1.5 kJ·g$^{-1}$ whereas the values ranged between −0.5 to −2.0 kJ·g$^{-1}$ for the other soils (see Supplementary Material Table S1 Sheet "TGA-DSC data"). The results regarding thermal recalcitrance of organic matter are shown in Figure 3 as the temperature at which 50% of SOM is combusted ($T_{50}$) pointing to significant differences between the soil samples ($p < 0.05$) with values ranging 330° to 350 °C. The highest $T_{50}$ were exhibited by the surface layers of *U. dioica*-colonized soil with a median of 341 °C, subsequently followed by the $T_{50}$ of bulk layers of *U. dioica*-colonized soil with a median of 338 °C and the surface layers of *I. glandulifera*-colonized soil with a median of 335 °C.

The lowest thermal resistance was found for the bulk and surface layers of the non-populated control soil with a $T_{50}$ with a median of 334 °C. Significant differences in $T_{50}$ were reported between bulk layers of soil colonized by *I. glandulifera* and surface layers colonized with *U. dioica* ($p = 0.01$) and between the bulk layers of the non-populated control soil and the surface layers of soil colonized with *U. dioica* ($p = 0.02$) Mean values and standard errors are provided in the Supplementary Material (see Table S1 sheet "TGA-DSC-data").

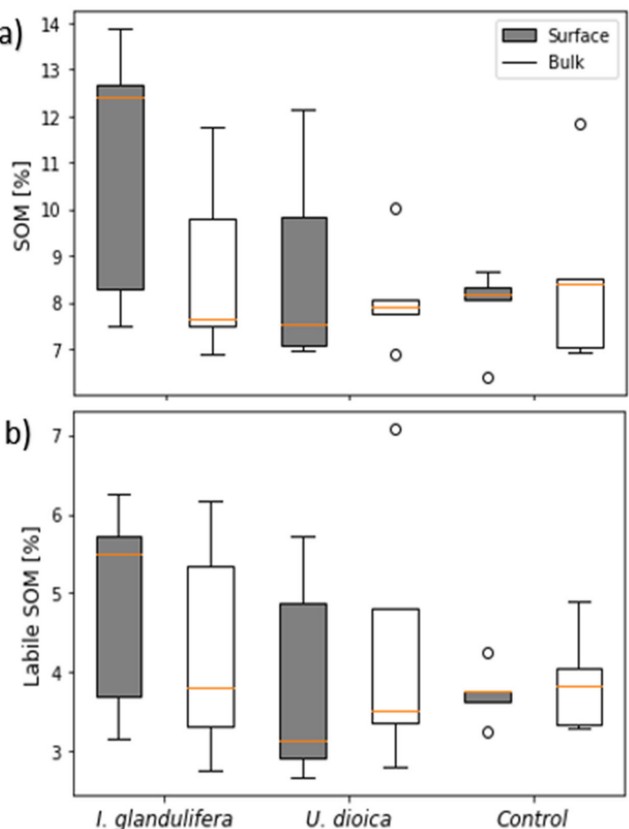

**Figure 2.** Percentage of (**a**) total soil organic matter (SOM) and (**b**) labile soil organic matter (Labile SOM) on a dry mass basis for surface (0–2 cm) and bulk layers (2–5 cm depth) of soil colonized by *Impatiens glandulifera* (*I. glandulifera*) and by *Urtica dioica* (*U. doica*) or of bare soil (Control) obtained as thermogravimetric mass loss between 200 and 600 °C. The boxes present the lower and upper 25th percentiles with whiskers showing the 1.5 fold interquartiles range (IQR) and all data points outside the 1.5 fold IQR depicted as empty circles.

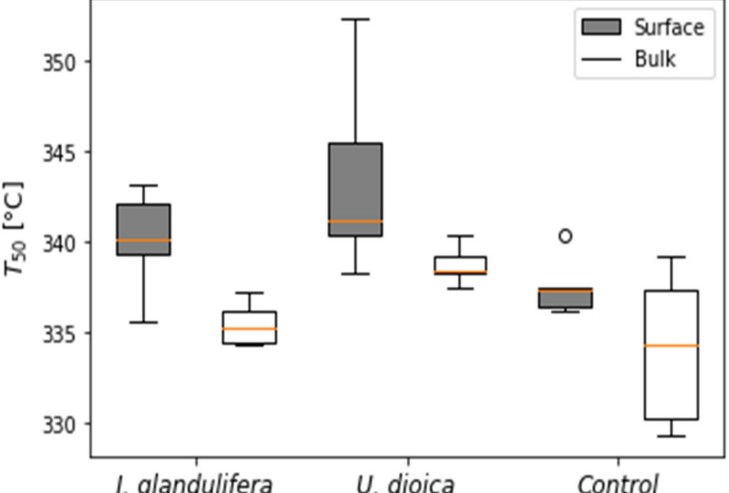

**Figure 3.** Thermal stability of SOM expressed as the temperature in °C at which 50% of the SOM mass was burnt (T50) for surface (0–2 cm) and bulk layers (2–5 cm depth) of soil colonized by *Impatiens glandulifera* (*I. glandulifera*) and by *Urtica dioica* (*U. doica*) or of bare soil (Control) obtained from thermogravimetric mass loss curves between 200 and 600 °C. The boxes present the lower and upper 25th percentiles with whiskers showing the 1.5 fold interquartiles range (IQR) and all data points outside the 1.5 fold IQR depicted as empty circles.

### 3.2. Soil Physico-Chemical Properties

The pH of the investigated soils ranged between 6–8 (Figure 4a). Control and *I. glandulifera* colonized soil showed neutral pH, which slightly decreased with soil depth. Soil colonized by *U. dioica* was more acidic and the pH decreased with soil depth as well and ranged between 6–7. Only significant difference in pH was found between surface layers of soil colonized with *U. dioica* and the bulk layers of the non-populated control soil which showed a slightly alkaline behavior ($p = 0.04$).

Soil depth and vegetation type showed no significant effects on the soil EC (Figure 4b). The EC ranged from 2–378 $\mu$S·cm$^{-1}$ but did not show a clear tendency between the soil depths nor the vegetation cover. Mean values and standard errors are provided in the Supplementary Material (see Table S1 sheet "pH, EC").

### 3.3. Soil Microstructural Stability

Results of the rheological measurements revealed for $\tau_{YP}$ and $\gamma_{(\tau max)}$ no statistically significant effects ($p < 0.05$) of soil depth or vegetation on the soil microstructural stability. Soil colonized by *I. glandulifera* showed a slight increase for $\tau_{YP}$ with soil depth while soil colonized by *U. dioica* and the controls showed the opposite trend with decreasing $\tau_{YP}$ with soil depth (Figure 5a).

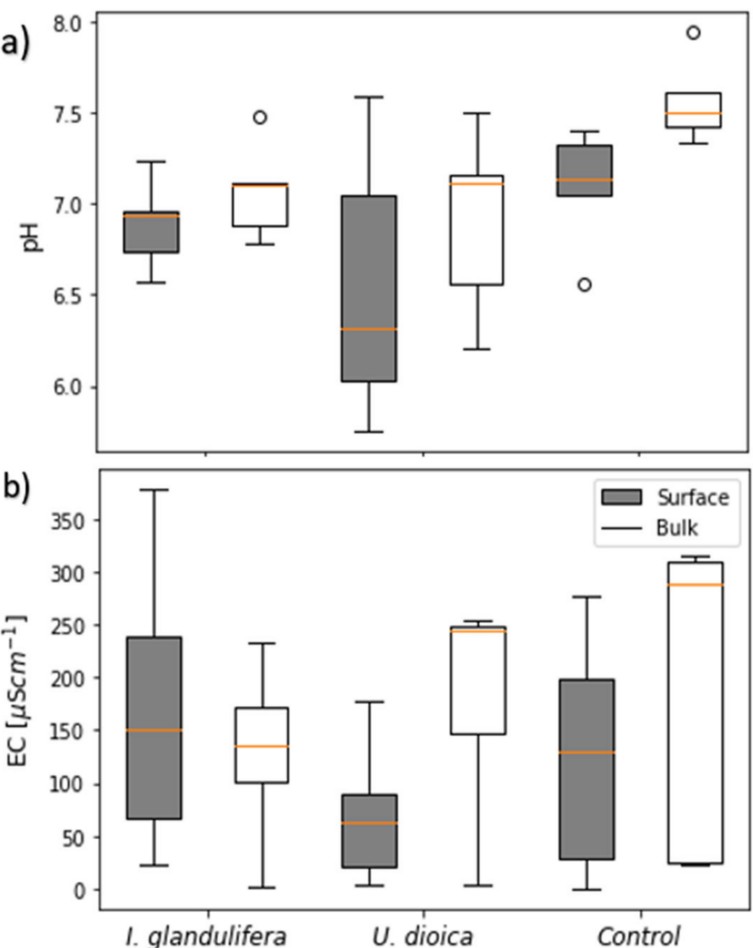

**Figure 4.** Physico-chemical soil parameters (**a**) pH value, and (**b**) electrical conductivity (EC) in $\mu$S cm$^{-1}$ for surface (0–2 cm) and bulk layers (2–5 cm depth) of soil colonized by *Impatiens glandulifera* (*I. glandulifera*) and by *Urtica dioica* (*U. doica*) or of bare soil (Control). The boxes present the lower and upper 25th percentiles with whiskers showing the 1.5 fold interquartiles range (IQR) and all data points outside the 1.5 fold IQR depicted as empty circles.

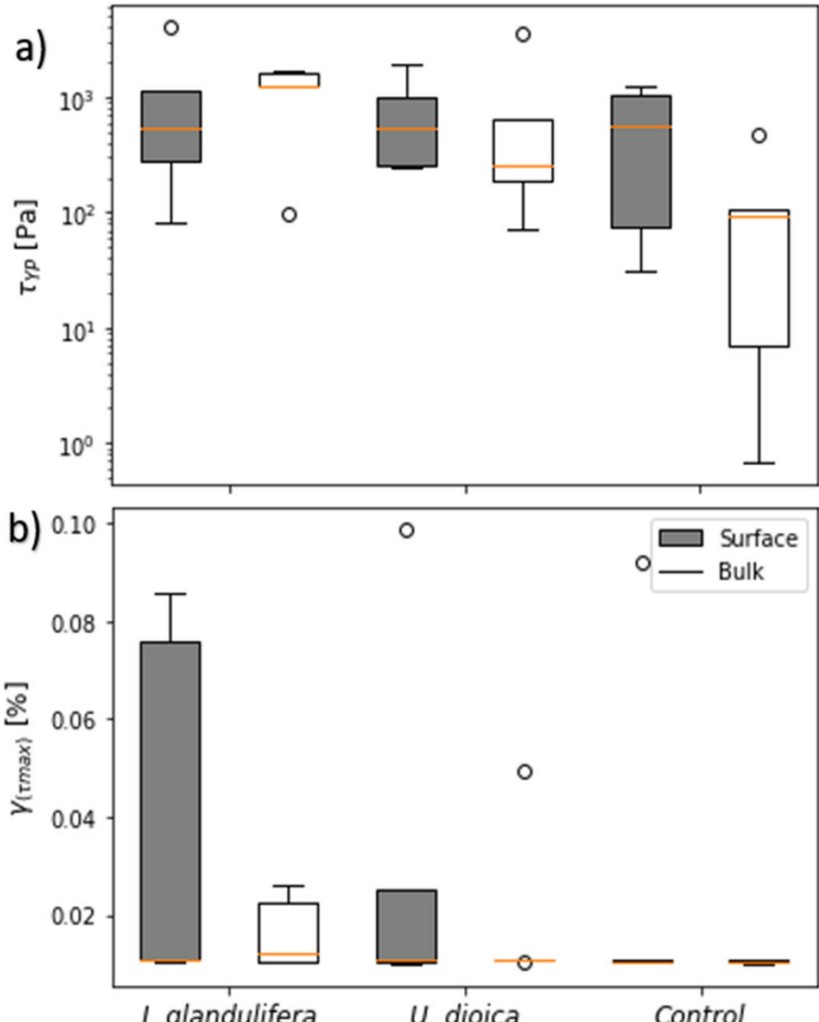

**Figure 5.** Microstructural stability expressed as (**a**) shear stress at the yield point (τYP) in Pa, and (**b**) corresponding strain to the maximum shear stress γτmax for surface (0–2 cm) and bulk layers (2–5 cm depth) of soil colonized by *Impatiens glandulifera* (*I. glandulifera*) and by *Urtica dioica* (*U. doica*) or of bare soil (Control). The boxes present the lower and upper 25th percentiles with whiskers showing the 1.5 fold interquartiles range (IQR) and all data points outside the 1.5 fold IQR depicted as empty circles.

Although the strain corresponding to the maximum shear stress $\gamma_{(\tau max)}$ did not differ in the median values, remarkable differences in the data scattering between soil depths and vegetation types were observed. The highest heterogeneity in $\gamma_{(\tau max)}$ appeared for the surface soil colonized by *I. glandulifera* followed by surface soil colonized by *U. dioica* and finally the bulk soil colonized by *I. glandulifera*. In contrast, $\gamma_{(\tau max)}$ values appeared most homogeneous in the bulk soil colonized by *U. dioica* and both layers of bare soil (control). Mean values and standard errors are provided in the Supplementary Material (see Table S1 sheet "Rheology-data").

### 3.4. Soil Hydraulic Properties and Initial Contact Angle

The pore size distribution (PSD) indicated that 97–98% of the water in all soils was located in the macropore domain (Figure 6). Only 1–2% of the water was located in micropores and about 0.1% in mesopores, respectively. The proportion of coarse pores in the field capacity ranged from 97.0–98.6% with neglectable differences among the soil depths and vegetation cover (Figure 6a).

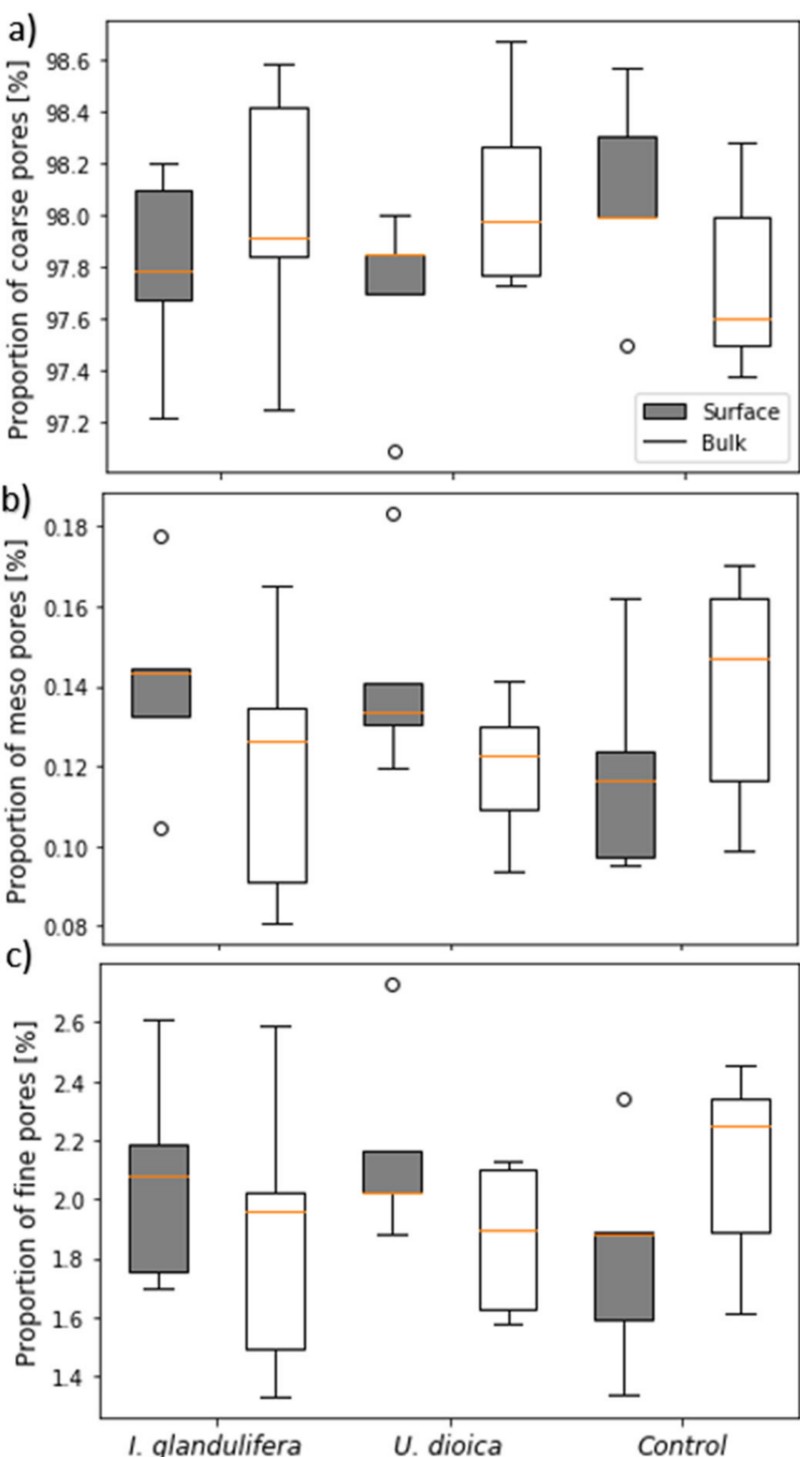

**Figure 6.** Pore size distribution as proportion of (**a**) coarse pores, (**b**) medium pores, and (**c**) fine pores at the field capacity from surface (0–2 cm) and bulk layers (2–5 cm depth) of soil colonized by *Impatiens glandulifera* (*I. glandulifera*) and by *Urtica dioica* (*U. doica*) or of bare soil (Control). The boxes present the lower and upper 25th percentiles with whiskers showing the 1.5 fold interquartiles range (IQR) and all data points outside the 1.5 fold IQR depicted as empty circles.

The proportion of mesopores was very low and did not show a clear tendency in relation to soil depth or vegetation type (Figure 6b). The proportion of the mesopores ranged between 1.4% and 2.6% while a clear trend was not detectable as well (Figure 6c). No Significant differences in the proportion of each pore domain between the vegetation

types were detected ($p > 0.05$). The water retention curves including the pore domain are included in the Supplementary Material (see Table S1 sheet "PSD").

The initial wetting of the investigated soils did not show significant differences. The bulk layers of soil colonized by *I. glandulifera* tended to have a higher initial contact angle of ~146° than the surface layers of the bare soil (control) and of the soil colonized by *U. dioica* with 86° and 112°, respectively (Figure 7). For the surface layers, soil colonized by *I. glandulifera* had a median of 132°, which was a comparable to the initial contact angle compared of soil colonized by *U. dioica* (median of 131°). However, the latter revealed a much lower variance. Despite the higher median of 146° for bare soil, differences in the initial contact angle between soils of different depths and vegetation types were not significant ($p > 0.05$). Mean values and standard errors are provided in the Supplementary Material (see Table S1 sheet "CA").

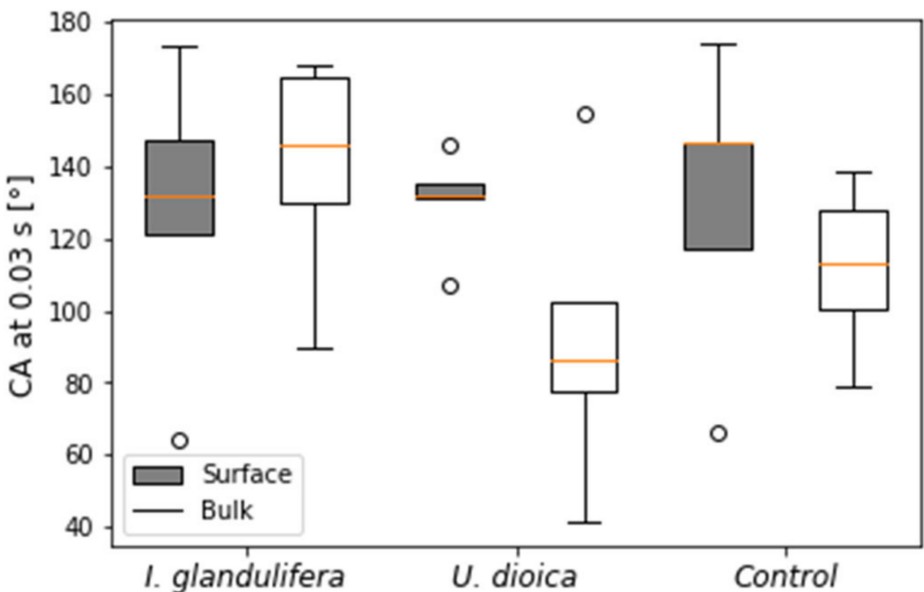

**Figure 7.** Initial wettability expressed as contact angle after 0.03 s contact time for surface (0–2 cm) and bulk layers (2–5 cm depth) of soil colonized by *Impatiens glandulifera* (*I. glandulifera*) and by *Urtica dioica* (*U. doica*) or of bare soil (Control). The boxes present the lower and upper 25th percentiles with whiskers showing the 1.5 fold interquartiles range (IQR) and all data points outside the 1.5 fold IQR depicted as empty circles.

### 3.5. Relationship between Soil Physico-Chemical Properties

Principal component analysis reduced the obtained parameters to ten principal components represented as arrows in Figure 8: SOM thermal properties (SOM content, $T_{50}$), soil structural stability indices ($\gamma_{YP}$ and $\gamma_{(\tau max)}$), soil hydraulic properties (percentage of the respective pore volumes, contact angle at 0.03 s) and general physico-chemical properties (pH, EC) to two principal components that explained >50% of the total variance. A clear clustering of the investigated soils could not be observed, however some relationships between soil properties were detectable. The first principal component PC1 that explained 33.9% of the sample variance most strongly relied on the proportion of mesopores (MP) and fine pores (FP) and negatively correlated with the proportion of coarse pores (CP). The latter was further strongly positively correlated with electrical conductivity (EC) that played a less dominant role in PC1. The second principal component PC2 explained 16.8% of the sample variances and most strongly relied on SOM content and pH, followed by the soil structure indices $\tau_{YP}$ and $\gamma_{(\tau max)}$ and the contact angle (CA) located in the second quadrant and thus correlated positively with each other.

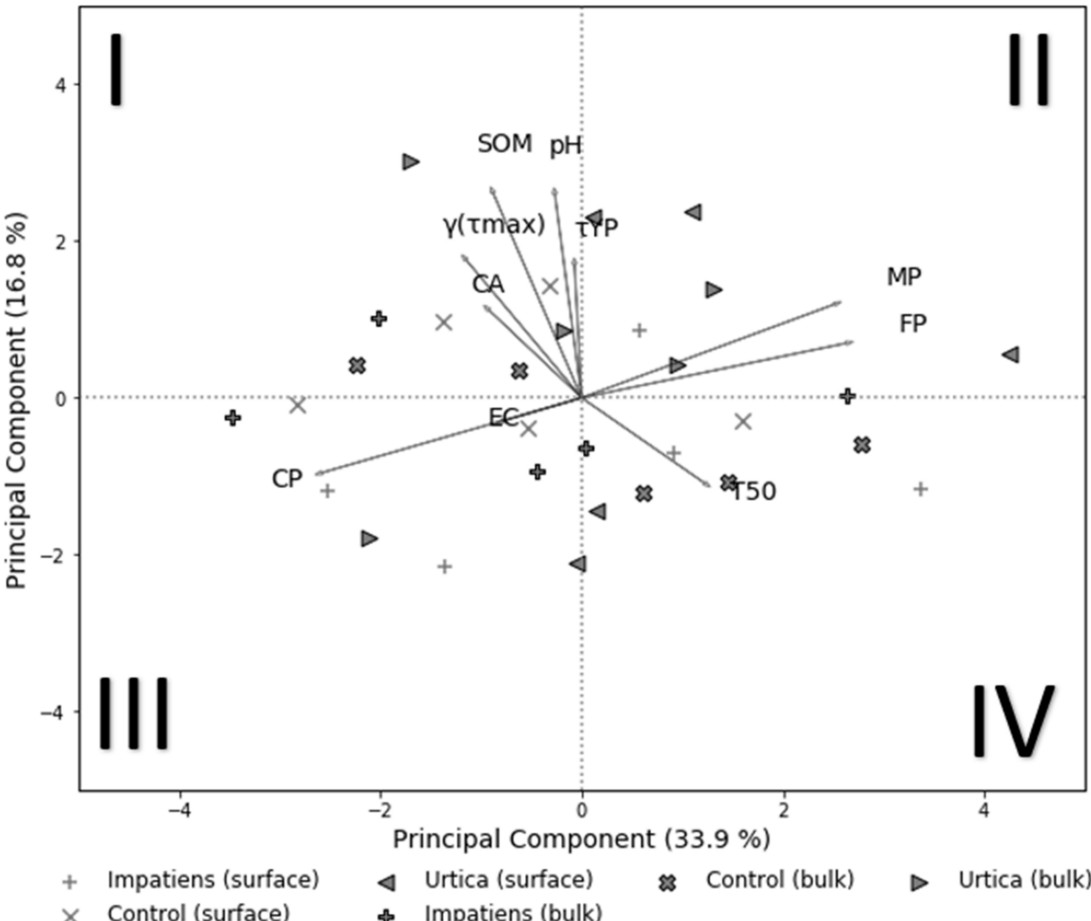

**Figure 8.** Principal Component Analysis (PCA) of parameters of surface (0–2 cm) and bulk layers (2–5 cm depth) of soil colonized by *Impatiens glandulifera* (*I. glandulifera*) and by *Urtica dioica* (*U. doica*) or of bare soil (Control). Displayed are relationships between total soil organic matter (SOM /%), thermal recalcitrance (T50/°C), soil hydraulic properties expressed as coarse (CP), medium (MP) and fine pores (FP/%) at the field capacity, wettability expressed as contaqct angle (CA/°), electrical conductivity (EC/µS·cm$^{-1}$), soil pH (pH), and soil structural stability expressed as shear stress at yield point (τYP/Pa) and shear strain at the end of the viscoelastic range (γ(τmax)/%). Quadrants of the projected coordinate system are given by roman numerals.

The T$_{50}$, located in the opposite quadrant IV was negatively correlated to the former parameter, with the strongest correlation to the CA. Furthermore, within the first two PCs (>50% of the variances), SOM content did not correlate to the any proportion of pores and also EC does not correlate neither to pH nor to the soil structure indices τ$_{YP}$ and γ$_{(τmax)}$. The respective principal components with their influence are provided in the Supplementary Material (see Table S1 sheet "PCA").

## 4. Discussion

### 4.1. SOM Fractions and Thermal Recalcitrance

#### 4.1.1. Depth and Plant Dependent Heterogeneity of SOM

Bare soil was not exposed to the same extend of root formation, neither transport of SOM nor formation or the release of organic root exudates compared to soil colonized by *I. glandulifera* or *U. dioica.* SOM mainly develops from plant and microbial biomass, as well from rhizodeposition [43] which is reflected in the lower values for bare soil with a lower input of fresh organic matter especially in the surface layers. SOM content reflects a higher heterogeneity and decreases with depth for the soil colonized by *Impatiens glandulifera* (invasive species). While soil colonized by *Urtica dioica* (native species) decreases in SOM

content and heterogeneity probably caused by the root architecture which might determine the spatial hot-spots for microbial degradation [44]. Microbial hotspots and a subsequent heterogenous formation of SOM depend strongly on plant roots has been reported to increase microbial activity 2-20 times compared to non-rooted soil [45]. The lateral roots of *U. dioica*, which typically roots in topsoil [13] spread closer along the surface while the tapered root of *I. glandulifera* reaches deeper into soil. This then results in hot-spots for microbial degradation at different depths where root exudates and the root itself could be directly metabolized by the soil microbial community.

### 4.1.2. Plant Dependent SOM Quality

Rather than in quantity, the effects of *I. glandulifera* were more pronounced in (Figure 2), the quality of SOM in (Figure 3) .

Biogeochemical stability [46] and to the degree of degradation [26] tended to be higher for Urtica-colonized soil. It also has been suggested, that the soil microbiota tends to mineralize organic matter with high energy content [47]. With an increasing degree of degradation SOM becomes more recalcitrant which might be reflected in a higher $T_{50}$ value [48]. However, the degradation process is complex and depends on several parameters such as the composition by the microbial community or the composition of plant tissue. Two main compounds originating from plant tissue were described to be cellulose and lignin which provide a carbon and an energy source for the soil microbiota [49]. Biopolymers like cellulose or hemicellulose are easier degradable and have been described as energy-yielding substrates, while lignin is more recalcitrant to degradation due to its more complex chemical bonds [50]. One reason for soil colonized by *U. dioica* exhibiting the highest $T_{50}$ might be the degree of decomposition [50,51]. For this species a litter cellulose content of around 86% has been reported [52], while stems from *I. glandulifera* mainly contain lignin and holocellulose [4,11]. A lignin content of approximately 50% for litter derived from *I. glandulifera* has been reported [53]. Therefore, we propose that the litter derived from *I. glandulifera* provides an energy intensive carbon source and most likely inhibit degradation by the soil microbiota resulting in a lower $T_{50}$ reflecting less degraded biomass. Another study suggested a lower bacterial activity due to colonization by *I. glandulifera* [54]. Allelopathic effects on the decomposer communities were described as well [8,17].

Additional studies focusing on microbial metabolism in different stages of vegetation or assessing the soil microbial activity can support this assumption.

### 4.2. Soil Structural Stability

Since no substantial differences depending soil depth nor vegetation cover were detectable we (Figure 5a) cannot confirm the hypothesis that colonization of soil by the native species improves soil microstructural stability. According to the PCA it is more likely that the microstructural stability is linked with SOM content, which is well-known [55]. Soil microstructural stability also depends strongly on soil texture. The interaction of soil minerals with organic matter [56] and is also expressed as the degree of aggregation [57]. The soil type investigated in this study contained a high share of sand, which is known to reduce soil structural stability [33,58]. At the same time the both clay and the SOM contents were relatively low and probably not sufficient to support the binding of soil particles [59,60]. Additionally studies imply that root hairs can reduce soil hardness and elasticity [61]. On a mechanistic scale, root hairs alter the soil structure by penetration [62]. With respect to our results, this might be reflected in a lower structural stability for soil colonized by *U. dioica*, where stability also decreases with soil depth (Figure 5)**.**

Soil colonized by *I. glandulifera* tended to show an enhanced density, probably caused by the root volume [63], which is concentrated in large tap root and different from the fine and lateral *Urtica* root. Our results indicate that both colonized soils exhibit a higher microstructural stability than the bare soil (control). Moreover, the bulk soils showed this

behavior and could imply an enhanced aggregation by plant roots however, the differences between the investigated root systems were only small.

### 4.3. Soil Hydraulic Properties

In general, the investigated soils were easily wetted independent of the vegetation type or differences in SOM content (Figure 7).

Thus, our hypothesis, that invasive plant colonization alters wettability by SOM input could not be confirmed. Our results indicated that the wettability of the investigated soils is mainly dominated by soil texture rather than SOM content. The investigated soils in our study showed large amounts of silt. Silty soils have been shown to a provide a good wettability [64]. Additionally, the quality of SOM given by the composition of organic molecules plays a key role in affecting the wettability [23], which was not assessed in this study but could be attempted in future attempts using analytical techniques such as Near-edge-x-ray absorption fine structure (NEXAFS) [65] in order to detect certain functional groups.

Since no significant differences were detected for the obtained PSD (Figure 6), we cannot confirm the hypothesis that swelling of SOM due to higher SOM input by *I. glandulifera* shifts the PSD towards smaller pores. In contrast, our results suggest that soil texture governs the development of PSD rather than the colonizing vegetation. The high percentage of macropores are caused by the large quantities of silt and sand. That almost no mesopores were detected implies a possible swelling of the silt fraction by which the mesopores may be strongly reduced in size and become micropores [25,66]. In agreement with our results, Marin et al. [61] studied the impact of root hairs on soil hydraulic properties in terms of sorptivity and repellency and could not report significant differences between root systems. Also no significant impact on water retention has been reported in a study investigating soil properties after invasive plant colonization by I. glandulifera [18]. However, another study suggests that coarse roots (>2 mm) and their decay promote the formation of macropores which enhances the permeability of soil but decrease the moisture hysteresis [67]. In accordance, we also detected a large share of macropores (>90%), but independent from the root architecture.

### 4.4. Soil Physico-Chemical Properties

The weakly acidic pH values of *U. dioica* colonized soil (Figure 4), especially in the surface layers are in line with other studies conducted in Europe [68]. Weakly acidic soils provide ideal growing conditions [13] due to the preference of nitrate as nitrogen donor [69,70]. Soils with a higher nitrification degree are slightly more acidic since H+-ions are released during ammonium oxidation [71]. We also observed, a more neutral pH for the bulk layers of soil colonized by U. dioica, where probably less roots are abundant. Soil colonized by Impatiens showed no impact on pH and is in line with former studies [18,72]. Also, the salinity showed no impact of vegetation. However, the lower EC values in the surface layers of *U. dioica*-colonized soil support an impact of the lateral root structure by a root accumulation in the surface layer and an enhanced uptake of nutrient ions [69,70]. Since we investigated soils from a floodplain area, the impact of flooding [73] caused by heavy rain events on soil pH and the EC are also important, since anoxic conditions can impact the soil pH. A lack of oxygen can subsequently force the soil microbiome to change to anaerobic metabolism [74] which is known to affect the soil pH [75]. Soil pH also controls the microbial activity and therefore degradation in soil [76]. Ideal conditions were reported 6.5 and 8.0 [77,78] which we also observed for the investigated soils in our study. Regarding the EC, which is linked to solved ions in the soil solution [79,80], rain or flooding might cause spatial variation of the EC [81]. However, our results most likely hint on a slight influence by the rhizome [76] of *U. dioica* while the values for *I. glandulifera* and bare soil showed no remarkable differences, especially for the soil pH. The distribution of the electric conductivity is heterogenous and did not show clear trend implying these heterogenous pattern for river banks [81].

*4.5. Relationships between Soil Properties*

No clear separation of the distribution of the sample points in the PCA plot indicates that neither the vegetation type nor soil depth showed remarkable differences (Figure 8). A similar observation has been made by the analysis of variance with no significant differences between vegetation type and depth for most variables. Apart from PSD and EC, most soil parameters were correlated with the SOM content. A close relationship between soil structural indices and the SOM content is in accordance with literature and is mainly explained by the interaction of organic substances with soil particles causing a spatial reorientation [34,82]. Especially during rewetting events, adsorption of hydrated and swollen SOM substances onto minerals [59] and specifically the interaction with the clay fraction could benefit the microstructural stability [56,83]. The PCA further indicate a close relationship of SOM content with the CA, which has been discussed in literature for similar findings [64,84]. It is striking that SOM content did not show any effect on the pore size distribution, which is contrary to former studies [25]. This underlines the assumption that the impact of soil texture dominates the PSD while the SOM content differences are too small to detect any significant effect on it. The $T_{50}$ showed a negative correlation with pH, CA, soil structural stability and SOM content. The correlation of $T_{50}$ with CA and soil structural stability can be explained by a co-correlation with SOM alone. A negative relationship of the $T_{50}$ with the soil pH, has been reported with a higher activation energy for heterotrophic respiration and reduced soil pH [85]. With respect to our results, we observed an enhanced $T_{50}$ values for soil colonized by *U. dioica* which also had more stable SOM at the same time and a lowered pH value especially in the bulk layers. To summarize, the PCA showed relationships between phyisco-chemical properties which are known and in line with the literature while the effect of the plant colonization was not significant for most variables.

## 5. Conclusions

Our case study revealed first insights into the question of how *Impatiens glandulifera* as an invasive plant species and its root architecture differently affects soils in floodplain landscapes in terms of biomass decomposition as well as various physicochemical properties compared to a native species or bare soil. The impact of invasive plant colonization on basic physico-chemical soil properties, wettability and soil hydraulic properties in both depths was weak. Sampling at the end of vegetation also allowed us to compare the impacts of both plant species after at least one vegetation period and how degradation by the soil microbial community might respond. However, sampling during spring and summer could provide a time dependent development of SOM content and quality. The study showed that the colonization of *I. glandulifera* mainly increases the proportion of the thermolabile SOM fraction whereas for the native plant species, *Urtica dioica*, the thermostable fraction was more dominant. It was suggested that litter of *I. glandulifera* is most likely mainly ligneous which has been reported to provide a less valuable nutrient source for microbes leading to the accumulation of less thermostable SOM. At the same time litter quality originating from *U. dioica* was described to mainly contain cellulose, which has been reported to be easier degradable. These results imply that future attempts should also pay more attention to microbial metabolism and possible shifts in microbial community structures, which could be achieved with respiration experiments or assessing different proxies related to microbial or fungal biomass.

**Supplementary Materials:** The following supporting information can be downloaded at: https://www.mdpi.com/article/10.3390/soilsystems6040093/s1, Table S1: SI_Jamin_et_al_2022_Case_Study.

**Author Contributions:** Conceptualization, J.J., C.B. and G.E.S.; methodology, C.B.; software, J.J.; validation, J.J., C.B. and D.D.; formal analysis, J.J., C.B., D.D., J.D. and G.E.S.; investigation, J.J., C.B. and M.M.; resources, J.J., C.B. and G.E.S.; data curation, J.J., C.B. and D.D.; writing—original draft preparation, J.J.; writing—review and editing, C.B., D.D., M.M. and J.D.; visualization, J.J.;

supervision, C.B.; project administration, G.E.S.; funding acquisition, G.E.S. All authors have read and agreed to the published version of the manuscript.

**Funding:** This study was funded by the German Research Foundation (DFG)–3326210499/GRK2360, the APC was funded by the environmental chemistry group of the University of Koblenz-Landau.

**Institutional Review Board Statement:** Not applicable.

**Informed Consent Statement:** Not applicable.

**Data Availability Statement:** See Supplementary Material.

**Acknowledgments:** We appreciated statistical advice by Jörg Rapp and Jakob Wolfram. Sincere thanks to Hermann F. Jungkunst, Katherine Munoz, Alessandro Manfrin, Alexis Peter Roodt, Kai Riess and Mirco Bundschuh comments on earlier versions of this manuscript. We thank Pradeep Chintipalli Krishna for laboratory assistance. We also thank the two anonymous reviewers for their impact on this manuscript!

**Conflicts of Interest:** The authors declare no conflict of interest.

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
