# Peer review of "Physico-Chemical Soil Properties Affected by Invasive Plants in Southwest Germany (Rhineland-Palatinate)—A Case Study"

_soilsystems, doi:10.3390/soilsystems6040093_

Round 1

Reviewer 1 Report

Comments in the text.

Reviewer 2 Report

Manuscript title:  Physico-Chemical Soil Properties Affected by Invasive Plants  in Southwest Germany (Rhineland-Palatinate) – A Case Study   

Manuscript id: soilsystems-2019629

Authors: Jamin et al.

The manuscript is particularly strong regarding the less studied topic and the experimental setup on soil characteristics under specific conditions. The manuscript regarding the topic and results presented is of interest to the environmental and soil science community and revisions based on the comments below are recommended before considering for publication.

Major comments

·       The reference numbers are mixed, in the text, you have used Author and year format, but in the reference list, they are numbered. Consider harmonizing them….

·       Line 52-58, Neither the aim nor hypothesis of the study is clear, and, the approach is -missing …. Maybe you need to condense the statement

·       Lake of scientific literature to support the statements and findings throughout the manuscript…... I have made some suggestions for that and more need it….

·       More information is needed for ALL Figure captions and define the abbreviation and units that are used. And adjust the significant figures for the table and manuscript.

Detailed comments:

Abstract

Line 13-17: A complicated sentence, please revise and check the grammar

Introduction:

Line 36-40: A reference is needed here.

Line 40-46: These are rather long sentences, better to break them down into more sentences.

In MM section

The soil is categorized as ‘slit loam ‘’ please correct it.

Table 1: It would be better to show the other soil characteristics, e.g organic content, Nitrogen content, pH, ………..

Literature references are missing for all sub-section. It would be better to cite the references that the procedure adopted.

Additional info is needed for the table caption, most importantly significant figures.

In MM section, what is the quality control (QC) data? There is no mention of the QC.

What is the accuracy of the instruments, recovery, LOD, and LOQ ……. These parameters are needed to report the efficiency of any analytical system.

In general, how many times you’ve recorded the data,? duplicate? Triplicate?..... what you mentioned in line 91 is not clear, please elaborate more on this

Line 117-122: this paragraph belongs to the introduction section, please consider rephrasing or moving the paragraph to the introduction.

R&D section

Figure 2-8: What are the boxes, the line inside the boxes, the whisker, and the circle represented in the figures, please explain. And also most importantly, how did you define the outliers?.

Line 424-438: Also consider discussing the impact of rain or water, and seasonal variation o the pH and EC, as they are highly relevant here.

Figure 7, you might not need this figure, as there is no difference between the samples, and you already mentioned that there is no significance between the samples – this optional.

Line 302: did you mean Figure 9, as there is no Figure 10?

These sections are repeating information already presented and explain things in an unnecessarily complicated way. The quality of the manuscript would benefit from the whole section being condensed, Line 282-310, line 343-373, line 424-438

Line 330-334: A reference is needed here

Line 397-401: A reference is needed here

Line 411-414: A reference is needed here, for example, you can use: https://doi.org/10.1016/j.scitotenv.2020.142822

Conclusion

Important conclusions! However, the future perspectives for the following research are highly crucial here …..

Round 2

Reviewer 2 Report

I am happy to see the manuscript improved nicely. The authors addressed all my comments adequately.

However to make sure the statements are supported by literature I will recommend adding citations in the following lines:

28:  https://doi.org/10.1038/s41598-020-76586-1

line 71:
https://doi.org/10.3390%2Fma14216658
https://doi.org/10.3390/land10121362
